# Ionic Liquid Electrolyte Technologies for High-Temperature Lithium Battery Systems

**DOI:** 10.3390/ijms26073430

**Published:** 2025-04-06

**Authors:** Eleonora De Santis, Annalisa Aurora, Sara Bergamasco, Antonio Rinaldi, Rodolfo Araneo, Giovanni Battista Appetecchi

**Affiliations:** 1Department of Chemical Engineering Materials Environment, La Sapienza University of Rome, Via Eudossiana 18, 00184 Rome, Italy; rodolfo.araneo@uniroma1.it; 2Italian National Agency for New Technologies, Energy and Sustainable Economic Development, Technologies and Devices for Electrochemical Storage (TERIN-DEC-ACEL) Technical Unit, Via Anguillarese 301, 00123 Rome, Italy; annalisa.aurora@enea.it (A.A.); antonio.rinaldi@enea.it (A.R.); 3Nanofaber srl, Via Anguillarese 301, 00123 Rome, Italy; sara.bergamasco@nanofaber.com

**Keywords:** phosphonium, imidazolium ionic liquids, high-temperature applications, high thermal stability, lithium batteries

## Abstract

The advent of the lithium-ion batteries (LIBs) has transformed the energy storage field, leading to significant advances in electronics and electric vehicles, which continuously demand more and more performant devices. However, commercial LIB systems are still far from satisfying applications operating in arduous conditions, such as temperatures exceeding 100 °C. For instance, safety issues, materials degradation, and toxic stem development, related to volatile, flammable organic electrolytes, and thermally unstable salts (LiPF_6_), limit the operative temperature of conventional lithium-ion batteries, which only occasionally can exceed 50–60 °C. To overcome this highly challenging drawback, the present study proposes advanced electrolyte technologies based on innovative, safer fluids such as ionic liquids (ILs). Among the IL families, we have selected ionic liquids based on tetrabutylphosphonium and 1-ethyl-3-methyl-imidazolium cations, coupled with per(fluoroalkylsulfonyl)imide anions, for standing out because of their remarkable thermal robustness. The thermal behaviour as well as the ion transport properties and electrochemical stability were investigated even in the presence of the lithium bis(trifluoromethylsulfonyl)imide salt. Conductivity measurements revealed very interesting ion transport properties already at 50 °C, with ion conduction values ranging from 10^−3^ and 10^−2^ S cm^−1^ levelled at 100 °C. Thermal robustness exceeding 150 °C was detected, in combination with anodic stability above 4.5 V at 100 °C. Preliminary cycling tests run on Li/LiFePO_4_ cells at 100 °C revealed promising performance, i.e., more than 94% of the theoretical capacity was delivered at a current rate of 0.5C. The obtained results make these innovative electrolyte formulations very promising candidates for high-temperature LIB applications and advanced energy storage systems.

## 1. Introduction

Lithium-ion technology has deeply revolutionized the energy storage market, dominating the electronic sector and making possible the diffusion of electric/hybrid auto-vehicles. However, the demand of highly performant devices is rapidly expanding, also requiring satisfying hard/challenging operative conditions. For instance, large-scale applications (particularly, deep-water drilling devices, gas/oil industry, but also stationary power sources, and automotive) require batteries to be able to safely operate even at high temperatures (around or above 100 °C), while maintaining acceptable performance and cycle life without significant degradation [1].

Nevertheless, at present, commercial Li-ion batteries (LIBs) are temperature-limited as they can only occasionally overcome 50–60 °C [1]. The presence of volatile and flammable organic electrolyte solvents can lead to a dangerous chain of events, such as overpressure, cell venting, burning, and explosion, with rapid cell dismantling [2]. In addition, the LiPF_6_ salt (generally used in standard LIB electrolytes) is thermally unstable and, in the presence of even moisture and/or oxygen traces, is able to generate toxic HF acid, thereby irreversibility ageing the electrochemical device and leading to cell performance decay [3,4]. Therefore, operative temperatures levelling or exceeding 100 °C are actually unthinkable for commercial LIB devices.

Electrochemical energy storage devices able to safely run in challenging thermal conditions are not commonly proposed because of the (*i*) narrow operational temperature range of electrolyte, (*ii*) instability at electrode/electrolyte interface, and (*iii*) safety risks of cell components [5,6,7]. The optimization and design of novel electrolyte architectures are essential, as well as understanding the influence of electrolyte behaviour and the role of electrolyte components on cell performance. Different approaches were proposed including concentrated electrolytes, incorporation of additives, and multi-salt and multi-solvent systems. However, in the frame of this scenario, it appears evident that an appealing approach for overcoming this drawback is the design of non-volatile, non-flammable, thermally robust electrolyte formulations capable of withstanding high operative temperatures.

Ionic liquids, which are salts molten below 100 °C (often at room temperature or below), were proposed as advanced electrolyte solvents for improving the safety and reliability of LIB devices [8,9] due to their appealing peculiarities (i.e., no measurable vapour pressure, remarkable flame-retardant characteristics, fast ion transport properties, high chemical/electrochemical/thermal stability, good power solvent). Phosphonium-based ionic liquids were found to exhibit higher thermal and electrochemical stability compared to those containing ammonium cations [10,11,12,13,14]. In the present work, attention was focused on the tetrabutylphosphonium (P_4444_)^+^ cation, which was selected because the steric hindrance and the symmetry of its structure are expected to allow high thermal and electrochemical robustness [10,11,12,13,14], although these factors do not promote the ion conduction and the low melting temperature. The (P_4444_)^+^ cation, commercially available as bromine salt (rather easy to be handled and purified with respect to alkyl-phosphines), was coupled with selected anions of the per(fluoroalkylsulfonyl)imide family for their appealing thermal/electrochemical stability and good transport properties [8]. In addition, we have investigated the 1-ethyl-3-methyl-imidazolium bis(trifluoromethylsulfonyl)imide (EMITFSI) ionic liquid, as previous results [8,9] revealed very good physicochemical and electrochemical properties, including thermal stability.

The (P_4444_)^+^-based and EMITFSI ionic liquids, synthesized and purified according to an eco-friendly procedure reported in detail elsewhere [15], were studied in terms of thermal behaviour, ion transport properties, and electrochemical stability even in the presence of the lithium bis(trifluoromethylsulfonyl)imide (LiTFSI) salt. LiTFSI was selected because of its favourable thermal/electrochemical stability and ion transport properties. In addition, LiTFSI was proved to form protective Al(TFSI)_3_ layer onto the cathode aluminum current collector avoiding corrosion phenomena [16]. Preliminary cycling tests were run at high temperature in lithium iron phosphate (LFP) cathodes for checking the feasibility of these electrolyte formulations.

## 2. Results and Discussion

The IL materials were successfully synthesized with purity level overcoming 99.9 wt.%, i.e., particularly, the bromide, moisture, and lithium contents were found to be below 5 ppm. In total, four ionic liquids were prepared and investigated as electrolyte components: tetra-butyl-phosphonium (trifluoromethylsulfonyl)(nonafluorobutylsulfonyl)imide (P_4444_IM_14_), tetra-butyl-phosphonium bis(trifluoromethylsulfonyl)imide (P_4444_TFSI), tetra-butyl-phosphonium (fluorosulfonyl)(trifluoromethylsulfonyl)imide (P_4444_FTFSI), and 1-ethyl-3-methyl-imidazolium bis(trifluoromethylsulfonyl)imide (EMITFSI). The (IM_14_)^-^ and (FTFSI)^-^ anions were selected because of their asymmetric structure with the aim of lowering the IL melting point [8,9]. The P_4444_TFSI and P_4444_FTFSI samples are solid at room temperature. Figure 1 shows a picture of the pure P_4444_TFSI sample (left panel) and its electrolyte mixture with LiTFSI (right panel, IL:LiTFSI mole ratio = 4:1).

### 2.1. Thermal Properties

Figure 2 compares the DSC (Differential Scanning Calorimetry) trace of the pristine IL materials and of their electrolyte mixtures with the LiTFSI salt. The pure ionic liquid samples show a well-defined melting feature (labelled with an asterisk in Figure 2) whereas additional endothermic peaks (not evidenced in the thermal trace of EMITFSI) are ascribable to solid–solid phase transitions prior to melting the IL sample, likely due to internal structural rearrangements [8,9], which seem to depend on the anion asymmetry. For instance, the P_4444_TFSI sample (i.e., containing the symmetric TFSI anion) shows three peaks (located at −70, 28, and 65 °C, respectively) whereas two features are detected in the DSC trace of P_4444_FTFSI (around −28 and 12 °C, respectively) and P_4444_IM_14_ (around −35.9 and 7.0 °C, respectively), which house an asymmetric anion (in particular, IM_14_), prior to melting.

The (P_4444_)^+^-based ionic liquids displays higher fusion temperature than EMITFSI (i.e., the melting peak onset is located around −7.5 °C), likely attributed to the large steric hindrance of the tetra-butyl-phosphonium cation that enhances the van der Waals interactions with the anion [8]. For instance, P_4444_TFSI, P_4444_FTFSI, and P_4444_IM_14_ exhibit melting feature onsets located at 84.5, 65, and 21.8 °C, respectively, i.e., the melting feature is seen to progressively shift down to lower temperatures when passing from P_4444_TFSI to P_4444_IM_14_, likely due to increasing asymmetry of the anion, which leads to more and more unfavourable ion packing [8]. Conversely, the highest melting point is displayed from the P_4444_TFSI sample attributed to the symmetric structure of both cation and anion, resulting in easier ion packing and, therefore, remarkable ion lattice energy value [8].

The addition of the LiTFSI salt (IL:LiTFSI mole ratio = 4:1) to the investigated ionic liquids results in different behaviour. For instance, the 0.8P_4444_TFSI-0.2LiTFSI and 0.8P_4444_FTFSI-0.2LiTFSI electrolytes exhibit a shift in the melting feature, similarly to the solid–solid phase transition profiles, towards lower temperatures (with respect to the neat IL material, as depicted in Figure 2), likely ascribable to the remarkably reduced steric hindrance of Li^+^ with respect to the IL cation, which hinders the crystal lattice formation [8]. In most of the IL families, this effect overcomes the increase in the cation–anion interactions, due to the higher surface charge density of the smaller lithium cation, resulting in melting temperature decrease [8]. It is also interesting to note that the endothermal features, i.e., ascribed to both melting and solid–solid transitions, appear less split with respect to those observed in the DSC trace of the neat ILs. Conversely, the 0.8P_4444_IM_14_-0.2LiTFSI and 0.8EMITFSI-0.2LiTFSI samples show no feature apart from the glass transition profile. For instance, no crystallization (endothermal) peak was evidenced even under slow cooling or repeated low temperature thermal cycling. This unexpected behaviour, however, recorded in other IL materials [17] is likely ascribable to much slower crystallization kinetics of the ionic liquid in the presence of the LiTFSI salt.

### 2.2. Ion Transport Properties

The ion transport properties of the neat ionic liquid materials (open squares) and their electrolyte mixtures (full squares) with the LiTFSI salt are reported in Figure 3, in terms of ionic conductivity vs. temperature dependence, as Arrhenius plots.

Conduction values lower than 10^−8^ S cm^−1^, i.e., not detectable through the used equipment, were not reported. At lower temperature, the pure (P_4444_)^+^ IL samples show a marked progressive conductivity increase prior to melting (less evidenced in EMITFSI), suggesting that the ions are able to move even if the samples are still in solid phase (e.g., as a result of substantial gained ion mobility prior to material being fully molten). This behaviour agrees with the DSC results (Figure 2), which display low temperature solid–solid phase transitions. The solid–liquid transition is revealed by a step conductivity jump up to four orders of magnitude, once more in good agreement with the thermal measurements. For instance, Figure 4 compares the DSC trace and the conductivity vs. temperature dependence of the neat ionic liquid materials. It is worthy noting the correspondence of the step conduction rises with the endothermal melting feature and the progressive conductivity increases, detected prior to melting, with the solid–solid phase transition peaks.

As clearly evidenced, the endothermal feature attributed to the melting of the IL sample occurs just in correspondence of the conductivity step rise, which can be ascribable to the solid–liquid phase transition. Such a behaviour was also observed in the other ILs investigated, even in the presence of the LiTFSI salt. However, despite a well-evidenced conduction jump, the 0.8EMITFSI-0.2LiTFSI and 0.8P_4444_IM_14_-0.2LiTFSI electrolytes did not show any feature except for the glass transition profile (Figure 2). This apparent discrepancy, previously observed in other IL materials [17], can be addressed to the different protocols followed for conductivity measurements (i.e., preliminary treatment in liquid nitrogen, overnight sample hosting at −40 °C, rough cell electrodes, much larger sample mass) with respect to that adopted for the DSC ones (see the Section 3), which allowed for full crystallization of both the neat IL and the IL-LiTFSI electrolyte mixtures. Further rise in temperature above the melting point results in a more moderate increase in conductivity, which exhibits a VTF behaviour previously encountered in other IL materials [8,17] and typical of amorphous electrolytes [18].

The incorporation of LiTFSI leads to, even if modest, conductivity decay (i.e., reported in the literature for various IL electrolyte families) in the molten state ascribable to the higher surface charge density of Li^+^ with respect to the IL cation, thus increasing the ion–ion interactions and, therefore, the viscosity of the IL material. More interestingly, the presence of the lithium salt is found to remarkably lower the melting temperature of the ionic liquid sample. For the sake of truth, this behaviour was previously observed in several ionic liquid typologies [8], however no IL class until now investigated has exhibited melting point decay exceeding 40 °C (0.8P_4444_TFSI-0.2LiTFSI) or even 60 °C (0.8P_4444_FTFSI-0.2LiTFSI). For instance, the P_4444_TFSI ionic liquid is solid at room temperature whereas the 0.8P_4444_TFSI-0.2LiTFSI electrolyte mixture is in the molten state (Figure 1). Such behaviour, in good agreement with the DSC results in Figure 2, has to be attributed (as discussed above) to the much more reduced steric hindrance of the Li^+^ cation with respect to the (P_4444_)^+^ one, which strongly hinders the formation of the IL crystal lattice (i.e., largely overcoming the increase in the ion–ion interactions due to the higher surface charge density of Li^+^ with respect to the IL cation) and, therefore, leading to fusion point decay. This remarkably extends the operative temperature range of the investigated ionic liquids.

Table 1 summarizes the conductivity values, recorded at selected temperatures, for both the neat IL materials and the 0.8IL-0.2LiTFSI electrolyte mixtures. As also evidenced from Figure 3, the (EMI)^+^ formulation exhibits the highest ion conduction in the whole temperature range investigated in conjunction with the widest molten state interval. This is mainly due to the reduced steric hindrance of the imidazolium cation, with respect to the (P_4444_)^+^ one, and its quasi-planar structure, which allows for faster mobility [8]. For instance, a conductivity exceeding 10^−4^ S cm^−1^ is recorded already at −20 °C, making the 0.8EMITFSI-0.2LiTFSI formulation potentially appealing also for low temperature applications, whereas 10^−3^ S cm^−1^ and 10^−2^ S cm^−1^ are overcome at 10 °C and 80 °C, respectively. The better ion transport properties of the 0.8EMITFSI-0.2LiTFSI electrolyte are also witnessed by its lower glass transition temperature (around −86 °C), as shown in Figure 2. Below 0 °C, the 0.8P_4444_IM_14_ formulation displays the highest conductivity values with respect to the other tetra-butyl-phosphonium electrolytes, likely attributed to the high asymmetry of the (IM_14_)^-^ anion, which hinders the ion packing and, consequently, lowers the melting temperature of the IL sample [8]. Conversely, the 0.8P_4444_TFSI-0.2LiTFSI and 0.8P_4444_FTFSI-0.2LiTFSI formulations show larger conductivity in the molten state, as the lower steric hindrance of the (TFSI)^-^ and (FTFSI)^-^ anions with respect to (IM_14_)^-^ results in enhanced ion mobility. For instance, conductivity values approaching 2 mS cm^−1^ and overcoming 4 mS cm^−1^ are recorded at 70 °C and 100 °C, i.e., more than two and three times higher, respectively, with respect to the 0.8P_4444_TFSI-0.2IM_14_ formulation.

### 2.3. Thermal Stability

Strong thermal robustness is a crucial property in view of application in high-temperature devices. Figure 5 reports on the variable-temperature TGA (Thermo-Gravimetrical Analysis) traces of the ionic liquid materials investigated, as measured in nitrogen atmosphere. The EMITFSI material exhibits no appreciable weight loss up to 300 °C, i.e., about 20 °C (P_4444_TFSI and P_4444_FTFSI) and 50 °C (P_4444_IM_14_) higher with respect to the (P_4444_)^+^-based samples. At higher temperatures, degradation of the IL sample takes place, resulting in almost full weight loss (i.e., suggesting complete volatilization of the investigated ILs) in a temperature range from 300 to 400–500 °C.

To investigate the thermal robustness under hard conditions, isothermal step TGA tests were also carried out. The results, depicted in Figure 6, indicate remarkable lower stability with respect to the data obtained from variable-temperature measurements (Figure 5). This behaviour, previously observed in several IL families [8,9,17], supports the better reliability of the isothermal tests. For instance, the (P_4444_)^+^-containing materials start to exhibit a minimal weight loss (however, below 0.4%) above 175 °C, whereas no appreciable weight variation is observed for EMITFSI up to 200 °C. The nature of the anion (i.e., at least, the typologies investigated in the present work) does not seem to remarkably affect the thermal stability of the tetra-butyl-phosphonium ionic liquids. To summarize, the reported results clearly highlight an excellent thermal robustness up to 150 °C for the investigated ionic liquids.

### 2.4. Electrochemical Stability

Electrochemical stability is one of the most important requirements for electrolyte formulations, especially for those addressed to devices operating at high temperatures and high voltages. This topic peculiarity was evaluated through repeated anodic cyclic voltammetry (CV) tests performed on Li/0.8IL-0.2LiTFSI/C cells at 110 °C. The CV profiles, depicted in Figure 7, reveal relatively low current densities (normalized with respect to the geometrical area of the carbon working electrode) in the first anodic scan (dotted trace). However, a remarkable current decay is recorded in the following cycles (solid traces) in combination with the absence of cathodic features. This indicates that the irreversible oxidation processes (occurring during the first anodic sweep) can be related to contaminants rather than to the ionic liquid materials and/or to the LiTFSI salt.

Table 2 compares the voltage at which the current flowing through the cell achieves 10 (V_10_) and 20 (V_20_) μA cm^−2^ during the third anodic scan, with the value (anodic limit voltage, V_L(An)_) obtained by fitting the step raise region of the CV traces (third anodic sweep) vs. the X axis. As clearly evidenced from Table 2, the 20 μA cm^−2^ threshold is achieved above 4.6 V for the 0.8P_4444_FTFSI-0.2LiTFSI electrolyte and is overcome above 4.3 V and 4.1 V for the 0.8P_4444_IM_14_-0.2LiTFSI and 0.8P_4444_TFSI-0.2LiTFSI formulations, respectively. Conversely, the 0.8EMITFSI-0.2LiTFSI electrolyte matches a current density of 20 μA cm^−2^ already at 3.06 V. The 0.8P_4444_FTFSI-0.2LiTFSI sample exhibits a current value even lower than 10 μA cm^−2^ (V_10_) up to 4.5 V, whereas this limit is reached above 4 V only by the 0.8P_4444_IM_14_-0.2LiTFSI formulation. Therefore, the anodic electrochemical robustness of the investigated 0.8IL-0.2LiTFSI electrolytes is seen to mainly depend on the nature of the anion [8,9].

From a more accurate examination of the data reported in Figure 7 and Table 2, it appears evident that the nature of the cation can also play a role in determining the oxidation stability. For instance, the 0.8P_4444_TFSI-0.2LiTFSI electrolyte exhibits higher electrochemical robustness than 0.8EMITFSI-0.2LiTFSI, likely highlighting the better stability of the phosphonium cation with respect to the imidazolium one and confirming the influence of the cation on IL oxidation processes. For instance, the electron-donor effect of the *n*-butyl chain [19] could stabilize the positive charge localized onto the P atom of the (P_4444_)^+^ cation; however, further investigation (out of the purpose of the present work) should be carried out for understanding this behaviour.

Finally, the anodic limit voltage (V_L(An)_ values in Table 2) was determined through fitting of the step current riise region of the CV traces (third anodic sweep of Figure 7) vs. the X axis. It should be noted, as clearly reported in Table 2, that the V_L(An)_ parameter always largely exceeds 4.6 V, i.e., from about 0.1 (0.8P_4444_FTFSI-0.2LiTFSI electrolyte) to 1.6 V (0.8EMITFSI-0.2LiTFSI) higher with respect to the V_10_ and V_20_ values (Table 2). However, the corresponding current flow recorded at the V_L(An)_ voltage ranges from 60 (0.8P_4444_FTFSI-0.2LiTFSI) to 120 (0.8P_4444_TFSI-0.2LiTFSI) μA cm^−2^, i.e., the oxidation of the electrolyte sample is not fully negligible. Therefore, this approach, even if widely adopted in the literature [8,9] for estimating the electrochemical stability of electrolytes, could not provide an optimal evaluation. Overall, at temperatures overcoming 100 °C, the (P_4444_)^+^-based electrolyte formulations show electrochemical stabilities against oxidation well above 4.5 V, whereas the EMITFSI electrolyte is electrochemically stable only up to 3 V.

### 2.5. Preliminary Tests on Battery

Preliminary charge–discharge cycling tests were performed at 100 °C on Li/LiFePO_4_ cells (operating up to 4 V) to check the feasibility of the developed ionic liquid electrolyte technologies for devices operating at high temperatures. The 0.8P_4444_TFSI-0.2LiTFSI formulation was selected as the electrolyte with the aim to validate the possibility to use solid ILs at room temperature. The results are plotted in Figure 8 as the voltage vs. capacity profile recorded at different current rates. A flat plateau (i.e., maintaining the same voltage during almost the entire charge/discharge step), typical of the Li+ insertion/de-insertion process into the LiFePO_4_ active material [20], is observed (in the 3.0–3.3 V range) at 0.1C (black traces), with very good reproducibility in the following cycles (i.e., the voltage profiles are practically overlapped). The increase in the current rate up to 0.5C results in a progressively higher slope of the voltage profile as well as increasing ohmic drop (as expected) likely due to enhanced diffusive phenomena within the electrolyte. A nominal capacity close to the theoretical value (170 mA h g^−1^) [20] is delivered at 0.1C, highlighting the good behaviour of the 0.8P_4444_TFSI-0.2LiTFSI electrolyte operating at 100 °C with LiFePO_4_ cathodes. Interestingly, a capacity of 160 mA h g^−1^ (>94% of the theoretical value) is still discharged at 0.5C, suggesting good capability up to medium current rates and good compatibility at the electrolyte/electrode interface.

These results, even if preliminary, give clear indication about promising performance of the investigated electrolyte technologies for difficult operating conditions (high operative temperature/voltage, cycling tests run at 100% of deep discharge) without any apparent degradation. Of course, further optimization is required at the level of electrode formulation and cell design. An extended investigation on the cycling behaviour in batteries for these ionic liquid electrolyte technologies will be the subject of a future paper.

## 3. Materials and Methods

### 3.1. Ionic Liquid Electrolytes

The (P_4444_)^+^-based and EMITFSI ionic liquids were synthesized and purified according to a greener procedure, reported in detail elsewhere [15], which uses water as the only processing solvent. Compared to conventional synthesis methods, which often use expensive and toxic organic solvents (ethyl acetate, acetone, dichloromethane, acetonitrile [21]), this approach offers several benefits, including lower production costs and reduced environmental impact. Furthermore, the water-based synthesis method is highly scalable and allows for the full recycling of reagents and by-products, making it especially advantageous for large-scale production and future industrial applications.

The precursors, i.e., tetrabutylphosphonium bromide (P_4444_Br, purity ≥ 98 wt.%) and 1-ethyl-3-methyl-imidazolium chloride (EMICl, ≥98 wt.%), were purchased from Sigma-Aldrich (St. Louis, MO, USA). The lithium bis(trifluoromethylsulfonyl)imide (LiTFSI, battery grade, >99.9 wt.%) and lithium (fluorosulfonyl)(trifluoromethylsulfonyl)imide (LiFTFSI, >98 wt.%) were purchased by 3M (Maplewood, MN, USA) and Provisco (Brno, Czech Republic), respectively. The (nonafluorobutanesulfonyl)(trifluoromethanesulfonyl)imide acid (HIM_14_) water solution (60 wt.%) was provided by 3M. The activated charcoal, used as the purifying sorbent, was purchased from Sigma-Aldrich and previously rinsed in deionized water to remove eventual contaminants adsorbed onto its surface. A Millipore ion-exchange resin deionizer was used to produce deionized water (resistivity > 18 MΩ cm). The dried ionic liquid samples were housed and handled in an argon-controlled atmosphere glovebox (Jacomex, Dagneux, France, oxygen and moisture content below 1 ppm), and can be summarized as follows: P_4444_IM_14_, P_4444_TFSI, P_4444_FTFSI, EMITFSI.

The ionic liquids, designed and synthesized in the frame of the present work, were subjected to quality control through Karl-Fisher (KF) moisture titration and X-ray fluorescence to verify the halide content. The KF titrations were performed by a Mettler-Toledo water titrator using IL samples, coming from the synthesis and drying routes, with a weight from 0.5 to 1.0 g. The X-ray fluorescence measurements were conducted by an EDX-720 Shimadzu energy-dispersive spectrometer using a rhodium foil as the X-ray source, which operates at an energy level of 15, for elements from sodium to titanium or 50 kV for titanium to uranium. Housed in a cylindrical polypropylene cuvette, the sample under investigation was sealed at one end with a circular Mylar polyester window (10 mm in diameter and 6 μm thick), secured by a suitable ring to allow the passage of radiation. Care was taken for housing the IL sample to ensure the window was compact and uniformly covered. A calibration curve was obtained using aqueous potassium bromide solutions at known Br^−^ concentrations (expressed in 10^−3^ mol L^−1^). The correlation between the count number and bromide concentration was linear, with a regression coefficient of 0.999 and an absolute error of 0.005, described by the equation *y* = 0.613 *x*, where *y* is the bromide concentration while x represents the count number. This equation was used to determine the bromide content in ionic liquid samples.

The IL electrolyte formulations were prepared by dissolving the appropriate amount of LiTFSI into the ionic liquid under magnetic stirring at room temperature. The IL:LiTFSI mole ratio was fixed equal to 4:1 [8].

### 3.2. Thermal Measurememnts

The thermal properties were examined through Differential Scanning Analysis (DSC) by a TA Instruments mod. DSC250 calorimeter with a flow rate of 50 mL min^−1^. The IL samples (from 5 to 10 mg) were housed (within the glove box) in aluminum sealed pans. The measurements were carried out (both on the neat ILs and on the IL:LiTFSI electrolyte formulations) in nitrogen atmosphere by running a cooling scan (10 °C min^−1^) from the ambient temperature down to −120 °C, followed by a heating scan (10 °C min^−1^) from −150 up to 100 °C. The IL samples, which are in molten state at room temperature, were subjected to additional thermal cycling (run from −120 °C from the onset of endothermal melting feature) to fully crystalize the material under investigation [22].

The thermal stability was studied by Thermo-Gravimetrical Analysis (TGA) through variable-temperature ramps and isothermal measurements. A preliminary heating scan is essential for approximating the degradation temperature range of the samples with unknown thermal stability. However, isothermal methods give more reliable results, especially during prolonged heating periods [8]. The TGA measurements were performed by a TA Instruments mod. SDT650 calorimeter system in nitrogen atmosphere with a flow rate of 100 mL min^−1^. The IL samples under test (5–10 mg) were loaded in smooth alumina pans. The variable-temperature thermal measurements were carried out by running a temperature heating scan at a rate of 10 °C min^−1^ up to 600 °C, whereas the isothermal analyses were run by subjecting the samples (introduced into the calorimeter at room temperature) to five successive step heating periods (each lasting 3 h) from 100 up to 200 °C.

### 3.3. Determination of the Ionic Conductivity

The ion transport properties of the ionic liquid electrolytes were investigated in terms of ionic conductivity vs. temperature dependence. The measurements were performed using an AMEL mod. A 160 conductivity-metre, over a temperature range from −40 to 100 °C, with a slow heating scan rate (≤1 °C h^−1^) was used to better observe any possible phase transitions. The ionic liquid samples, handled within the glovebox, were housed into two-electrode, porous platinum cells which were immersed in liquid nitrogen for approximately one minute. Then, the cells were immediately transferred into a climatic chamber (Binder) set at −40 °C, for allowing full crystallization of the IL under investigation [22]. The liquid nitrogen treatment was repeated until the frozen ionic liquid samples remained solid at −40 °C. Subsequently, the cells were kept in the climatic chamber at −40 °C overnight before initiating the conductivity measurements. The specific ionic conductivity (*σ_sp_*), normalized with respect to the cell geometry, was determined using the equation $\sigma_{SP}=\sigma K$ where *σ* (S) represents the conduction value directly measured by the conductivity-metre and *K* (cm^−1^) is the cell constant (experimentally determined by a KCl aqueous solution at exactly known conductivity).

### 3.4. Anodic Cyclic Voltammetries

The electrochemical stability of the 0.2LiTFSI-0.8IL electrolytes was evaluated through anodic cyclic voltammetry (CV) run on Li/C cells using carbon-based working electrodes because they allow for better simulating the behaviour in a practical device. Conversely, inert working electrodes (such as aluminum, platinum, nickel), even if reported in the literature [23], represent only an ideal situation and do not reproduce the behaviour in a real device, i.e., they can lead to overestimation of the electrochemical stability. Therefore, aiming to obtain more reliable results, the electrochemical stability was investigated in carbon- and binder-containing working electrodes, with the target better approaching the cell conditions. Lithium metal was used as the counter electrode. The electrodes were prepared by blending 70 wt.% Super C45 (IMERYS) and 30 wt.% Na-carboxymethylcellulose (CMC, Dow Wolff Cellulosics) in deionized water, casting the so-obtained slurry onto aluminum foils (Honjo, 20 μm thick battery grade), and massively removing the aqueous solvent under a hood. Discs with diameters of 10 mm were punched from the electrode, then vacuum-dried at 150 °C overnight before being transferred within the glovebox. The Li/C cells, as well as the Li/LFP cells manufactured for the cycling tests, were fabricated by overlapping a lithium metal coin (10 mm diameter), a glass-fibre separator (16 mm diameter), and the carbon working electrode (or an LFP cathode). The IL electrolyte was spread onto both the working electrode (20 µL) and the separator (80 µL) and, successively, the so-obtained cell was vacuum-treated for 30 min before sealing for ensuring proper IL electrolyte penetration within the electrode/separator pores. Finally, the cells were housed in 2032 coin-cell containers (MTI) and sealed by a proper crimper (MTI).

The CV measurements were performed, using a galvanostat/potentiostat BioLogic, by consecutively scanning (1 mV s^−1^) the cell voltage within the 3.0–4.7 V range. The electrochemical tests were conducted in a climatic chamber (Binder) kept at 110 °C. To ensure reproducibility of the results, clean electrodes and fresh IL electrolytes were used for each test, and the experiments were repeated across different cells.

### 3.5. Charge-Discharge Cycling Tests 

The LFP cathodes were prepared by spreading a slurry containing the active material (Sud-Chemie, 85 wt.%), Super C65 carbon (IMERYS, 10 wt.%) as the electronic conducting additive, and polyvinylidene-difluoride (PVdF, Solef 6030 Solvay, 5 wt.%) as the binder, onto aluminum foil (Honjo, 20 μm thick battery grade). After massive solvent removal (within a hood), disc electrodes with diameters of 10 mm were punched, dried under dynamic vacuum (100 °C), and then pressed at 10-ton cm^−2^. The cathode mass loading approaches 3.4 mg cm^−2^, which, accounting for the LFP specific capacity equal to 170 mA h g^−1^, corresponds to 0.58 mA h cm^−2^. The Li/LFP half-cells (manufactured inside the glovebox) were subjected to galvanostatic cycling tests run, within the 2.0–4.0 V range at different current rates by a BioLogic multichannel battery cycler, at 100 °C (Binder climatic chamber).

## 4. Conclusions

Advanced electrolyte technologies, based on tetrabutylphosphonium and ethylmethylimidazolium ionic liquids (ILs), were formulated for being addressed to high-temperature lithium batteries. The so-prepared IL materials, i.e., P_4444_IM_14_, P_4444_TFSI, P_4444_FTFSI, and EMITFSI, were synthesized through a greener route, which allowed for lithium, halide, and moisture contents lower than 5 ppm. Only water was used as the processing solvent, representing a remarkable gain in terms of environmental impact and final cost. The electrolytic formulations were obtained by combining the IL materials with the LiTFSI salt (IL-LiTFSI mole ratio = 4:1).

Conductivity measurements, in good agreement with the DSC results, revealed fast ion transport properties already at 50 °C, particularly for the EMITFSI electrolyte. Ion conduction values ranging from 10^−3^ and 10^−2^ S cm^−1^ are levelled at 100 °C, indicating the feasibility to deliver large capacity even at high current rates. Interestingly, very good thermal robustness (above 150 °C) was recorded even under prolonged Thermo-Gravimetrical tests, in combination with stability against oxidation, investigated at 100 °C, exceeding 4.5 V. These topic peculiarities disclose the possibility of realizing lithium battery systems capable of safely and reliably operating at high temperatures, thus satisfying the challenging conditions that, until now, have not been matched by commercial LIB devices.

These very appealing perspectives are supported by preliminary cycling tests performed on Li/LiFePO_4_ cells at 100 °C, which delivered above 94 % of theoretical capacity at a current rate of 0.5C, making these electrolyte technologies particularly reliable for battery systems operating under challenging conditions.

In progress experimental activities foresee an investigation of the electrolyte/electrode interphase and the lithiation process as well as the optimization of the cathode formulation and cell design. These will be the subject of a future paper.

## Figures and Tables

**Figure 1 ijms-26-03430-f001:**
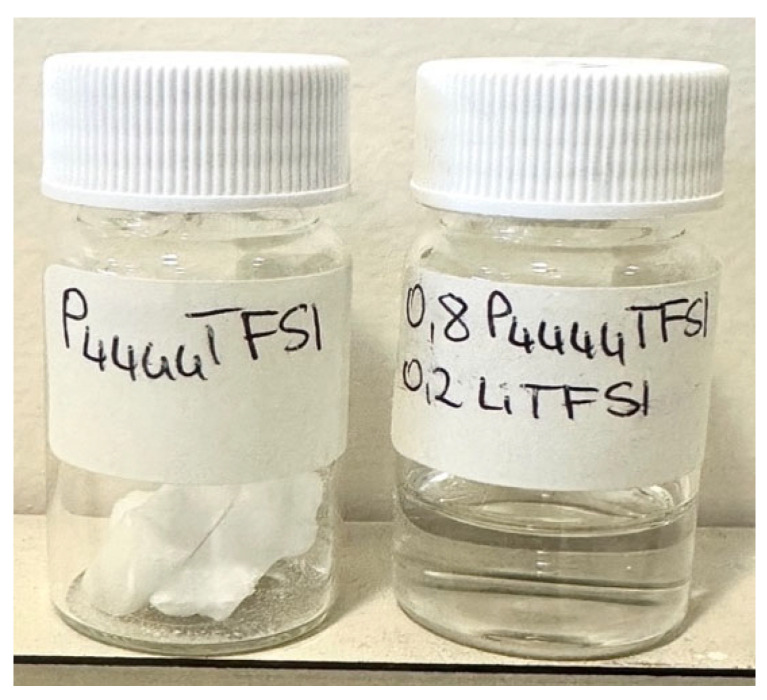
Pure P_4444_TFSI (**left**) and 0.8P_4444_TFSI-0.2LiTFSI (**right**) electrolyte formulation.

**Figure 2 ijms-26-03430-f002:**
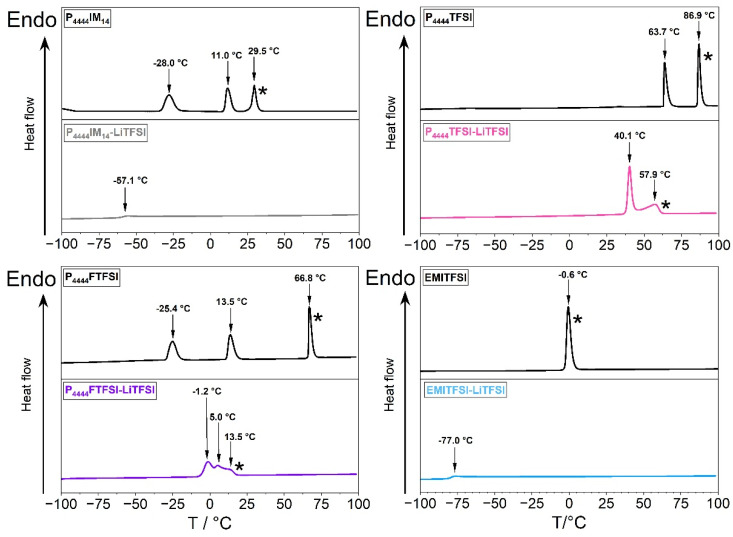
DSC trace of neat IL samples and 0.8IL-0.2LiTFSI electrolyte formulations. Heating scan rate: 10 °C min^-1^. IL:LiTFSI mole ratio = 4:1. Melting feature is highlighted with asterisk (*). Temperature value, at which endothermal phase transitions are located, are reported in panels.

**Figure 3 ijms-26-03430-f003:**
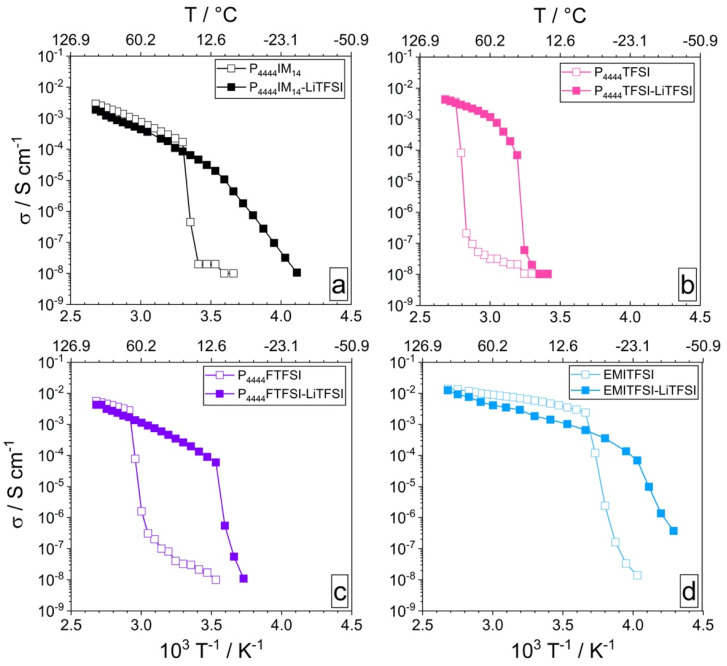
Ionic conductivity vs. temperature dependence, reported as Arrhenius plots, of investigated ionic liquids (open data markets) and their electrolyte mixtures (full data markets) with LiTFSI lithium salt (IL:LiTFSI mole ratio = 4:1). Heating rate: 1 °C h^−1^.

**Figure 4 ijms-26-03430-f004:**
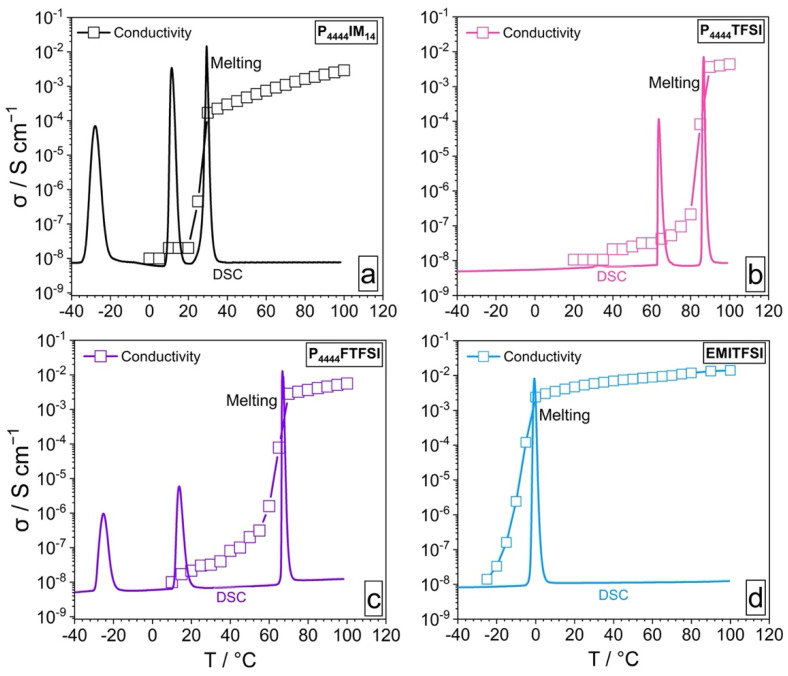
Ionic conductivity vs. temperature dependence and DSC trace of neat ionic liquid materials investigated in present work.

**Figure 5 ijms-26-03430-f005:**
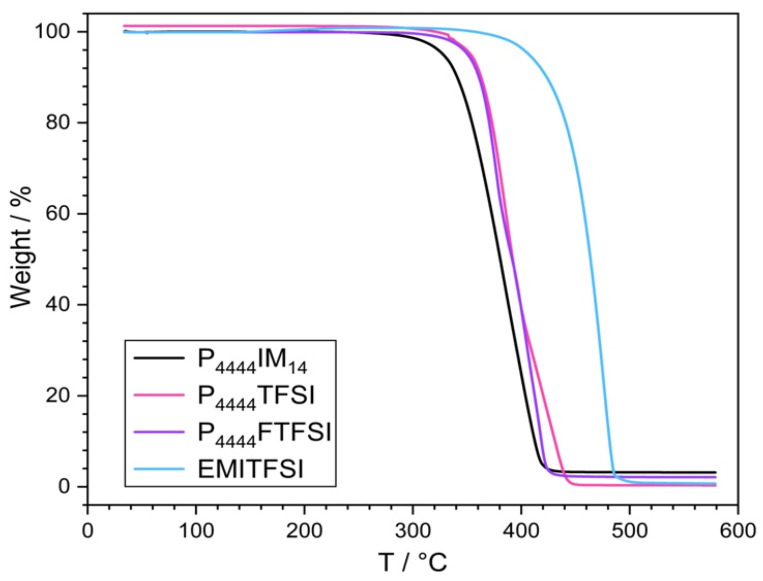
Variable-temperature TGA trace of investigated ionic liquids. Scan rate: 10 °C min^−1^.

**Figure 6 ijms-26-03430-f006:**
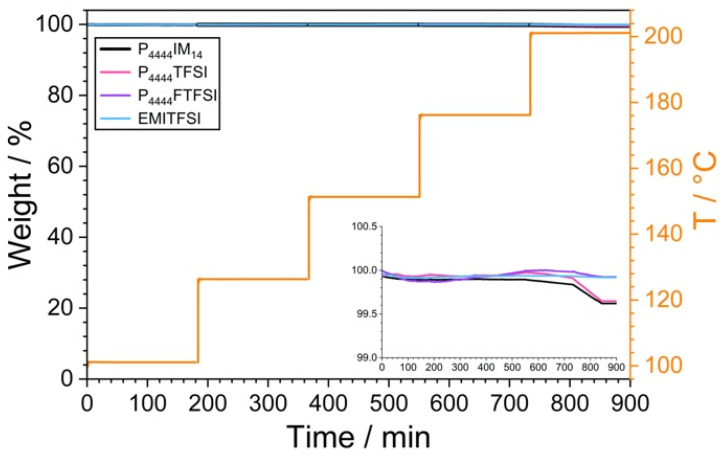
Isothermal TGA trace of the investigated ionic liquids. The measurements were run in nitrogen atmosphere. A magnification of the TGA curve is reported in the insert.

**Figure 7 ijms-26-03430-f007:**
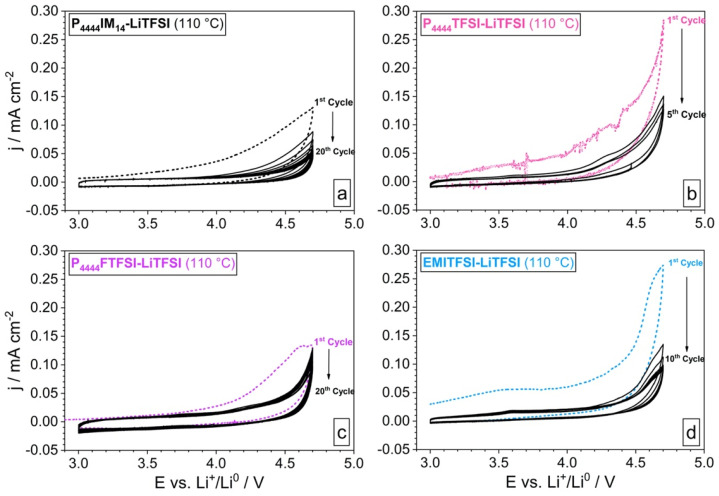
CV trace of investigated 0.8IL-0.2LiTFSI electrolyte formulations on carbon working electrodes. Lithium as counter. Scan rate: 1 mV s^−1^. Temperature: 110 °C.

**Figure 8 ijms-26-03430-f008:**
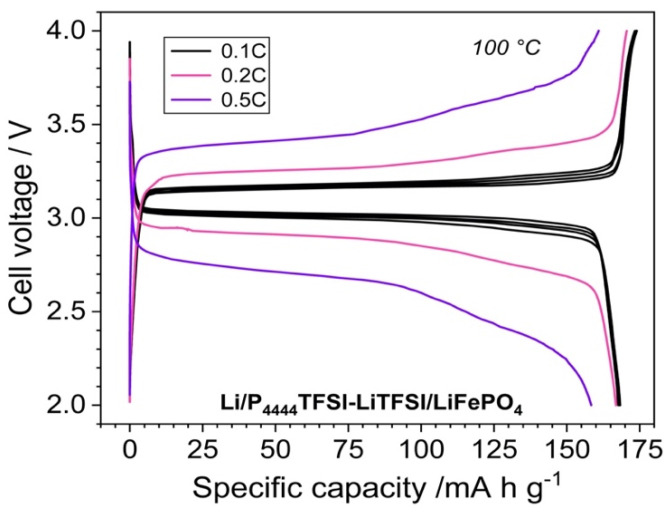
Selected voltage vs. capacity charge/discharge profiles, recorded at different current rates, of Li/0.8P_4444_TFSI-0.2LiTFSI/LiFePO_4_ cell. Temperature: 100 °C.

**Table 1 ijms-26-03430-t001:** Selected ionic conductivity values, determined at different temperatures, of investigated ionic liquids and their electrolyte mixtures with LiTFSI lithium salt (IL:LiTFSI mole ratio = 4:1).

Ionic Liquid Sample	Ionic Conductivity/S cm^−1^			
	0 °C	30 °C	70 °C	100 °C
P_4444_ IM_14_	(1.0 ± 0.1) × 10^−8^	(1.7 ± 0.2) × 10^−4^	(1.1 ± 0.1) × 10^−3^	(2.9 ± 0.3) × 10^−3^
P_4444_ IM_14_-LiTFSI	(4.4 ± 0.5) × 10^−6^	(8.5 ± 0.9) × 10^−5^	(6.3 ± 0.6) × 10^−4^	(1.9 ± 0.2) × 10^−3^
P_4444_TFSI	(1.0 ± 0.1) × 10^−8^	(1.1 ± 0.1) × 10^−8^	(5.3 ± 0.5) × 10^−8^	(4.4 ± 0.4) × 10^−3^
P_4444_TFSI-LiTFSI	(1.0 ± 0.1) × 10^−8^	(2.0 ± 0.2) × 10^−8^	(1.8 ± 0.2) × 10^−3^	(4.2 ± 0.4) × 10^−3^
P_4444_FTFSI	(1.0 ± 0.1) × 10^−8^	(3.2 ± 0.3) × 10^−8^	(2.8 ± 0.3) × 10^−3^	(5.6 ± 0.6) × 10^−3^
P_4444_FTFSI-LiTFSI	(5.5 ± 0.6) × 10^−8^	(2.6 ± 0.3) × 10^−4^	(1.7 ± 0.2) × 10^−3^	(4.3 ± 0.4) × 10^−3^
EMITFSI	(2.4 ± 0.3) × 10^−3^	(5.9 ± 0.6) × 10^−3^	(10 ± 1) × 10^−3^	(14 ± 1) × 10^−3^
EMITFSI-LiTFSI	(6.6 ± 0.7) × 10^−4^	(1.8 ± 0.2) × 10^−3^	(5.3 ± 0.5) × 10^−3^	(13 ± 1) × 10^−3^

**Table 2 ijms-26-03430-t002:** Voltage values, recorded in CV tests carried out on Li/IL:LiTFSI/C cells, when the current density achieves 10 μA cm^−2^ (V_10_) and 20 μA cm^−2^ (V_20_), respectively, during the third anodic sweep. The V_L(An)_ value represents the anodic limit voltage obtained by fitting the step current raise region of the CV traces (third anodic sweep) vs. the X axis. IL:LiTFSI mole ratio = 4:1. Temperature: 110 °C.

Ionic Liquid Electrolytes	Anodic Voltage Value/V		
	V_10_	V_20_	V_L(An)_
P_4444_IM_14_-LiTFSI	4.036 ± 0.001	4.319 ± 0.001	4.693 ± 0.001
P_4444_TFSI-LiTFSI	3.841 ± 0.001	4.135 ± 0.001	4.678 ± 0.001
P_4444_FTFSI-LiTFSI	4.562 ± 0.001	4.625 ± 0.001	4.649 ± 0.001
EMITFSI-LiTFSI	3.002 ± 0.001	3.055 ± 0.001	4.681 ± 0.001

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
