# Peer review of "Ionic Liquid Electrolyte Technologies for High-Temperature Lithium Battery Systems"

_ijms, 2025, doi:10.3390/ijms26073430_

Round 1
Reviewer 1 Report
Comments and Suggestions for Authors
The abstract should entice readers to engage with the paper. The necessity and objectives of the research are well stated. However, the results are missing. Summarize the results in one sentence and add them to the abstract. This will improve its quality.
In the introduction, include previous research conducted by other scholars. Clearly state how this study differs from previous works. Explicitly highlighting these differences will help establish the originality of this research.
Where is the experimental method described? Add a section detailing the experimental methods.
DSC is first mentioned in Section 2.1. What does DSC stand for? Is it Differential Scanning Calorimetry? Such details should be included in the experimental methods section. Revise the structure of the paper and clarify technical terminology.
According to the DSC results, an exothermic reaction occurs at a specific temperature. Was any crystalline phase or other material formed during this reaction? If a phase analysis was conducted after the exothermic reaction, include the results.
Figure 2 represents the heat flow as a function of temperature. Temperature is a critical factor in Li-ion batteries. However, the figure only shows temperature variations. Why are the temperature values not indicated? Add the temperature values.
TGA is similar to DSC. It first appears in Section 3.2 and is written as an abbreviation. Correctly define and present this term.
Why is the experimental method in Section 3? Organizing the paper in a logical sequence will help readers understand it better. Adjust the order of the experimental methods and results accordingly.
The conclusion should consist of both results and discussion. However, this conclusion merely lists the results, making it inadequate. Revise the conclusion based on references from previous literature.
Comments on the Quality of English LanguageThe English language should also be improved. Clearly articulate the objectives and results.
Author Response
Replies to Reviewer 1
The abstract should entice readers to engage with the paper. The necessity and objectives of the research are well stated. However, the results are missing. Summarize the results in one sentence and add them to the abstract. This will improve its quality.
Authors’ reply
The authors are grateful to the reviewer for the positive comments devoted to improving the manuscript quality. Changes in the text are highlighted in yellow.
The abstract was modified accordingly to the reviewer recommendations.
In the introduction, include previous research conducted by other scholars. Clearly state how this study differs from previous works. Explicitly highlighting these differences will help establish the originality of this research.
Authors’ reply
Electrolyte technologies able of stably operating at high temperatures are not often reported. However, the introduction was modified accordingly to the reviewer recommendations.
Where is the experimental method described? Add a section detailing the experimental methods.
Authors’ reply
The experimental methods are described in detail in Section 3 (Materials and Methods).
DSC is first mentioned in Section 2.1. What does DSC stand for? Is it Differential Scanning Calorimetry? Such details should be included in the experimental methods section. Revise the structure of the paper and clarify technical terminology.
Authors’ reply
The DSC acronym is widely used in literature and wouldn’t need to be clarified. However, the DSC acronym was clarified through the text.
According to the DSC results, an exothermic reaction occurs at a specific temperature. Was any crystalline phase or other material formed during this reaction? If a phase analysis was conducted after the exothermic reaction, include the results.
Authors’ reply
No exothermic peak is displayed in the DSC results of Figure 2, but only endothermal features. These are related to the melting of the sample under investigation or solid-solid phase transitions occurring prior the melting. However, we understand that the Y-axis notation of Figure 2 could lead to misleading conclusions, so it was modified.
Figure 2 represents the heat flow as a function of temperature. Temperature is a critical factor in Li-ion batteries. However, the figure only shows temperature variations. Why are the temperature values not indicated? Add the temperature values.
Authors reply
Frankly speaking, we did not understand what the reviewer does mean. If he/she means the temperature values at which the thermal transitions take place, we can state that they do not provide relevant added value to the Discussion and, therefore, can be omitted in the Figure. They are discussed in text.
TGA is similar to DSC. It first appears in Section 3.2 and is written as an abbreviation. Correctly define and present this term.
Authors’ reply
AS well as for DSC, the TGA acronym is widely used in literature and would not need to be clarified. However, the TGA acronym was clarified through the text.
Why is the experimental method in Section 3? Organizing the paper in a logical sequence will help readers understand it better. Adjust the order of the experimental methods and results accordingly.
Authors’ reply
The authors have followed the format of the Journal.
The conclusion should consist of both results and discussion. However, this conclusion merely lists the results, making it inadequate. Revise the conclusion based on references from previous literature.
Authors’ reply
The authors do not agree with the reviewer. As commonly reported in scientific Journals, the Conclusions are substantially a summary of the main achieved results as they are previously commented and discussed in detail in the Section Results and Discussion.
Reviewer 2 Report
Comments and Suggestions for Authors
In this manuscript, the authors explore the application of novel ionic liquids for high-temperature lithium batteries. The study thoroughly analyzes key properties, including melting point, ionic conductivity, and both thermal and electrochemical stability. Preliminary battery performance results indicate compatibility with commercial cathode materials. This work offers valuable insights into the potential use of ionic liquids as electrolytes for next-generation lithium batteries. I recommend the manuscript for publication after addressing the following concerns. Detailed comments are provided below.
- In Figure 2, the exothermic notation on the y-axis is misleading, as both endothermic peaks are observed for both IL and IL-LiTFSI.
- Although the ionic liquid systems show high ionic conductivity under high temperatures, one potential concern is the low Li transference number, which limits the Li+ transfer in batteries. It is recommended to provide a brief comment on the ion selectivity in IL systems. Additionally, is LiFSI salt a better choice considering the weaker binding of the FSI-?
- The EMITFSI system shows the lowest electrochemical stability, which appears inconsistent with previous studies. I suggest the authors use aluminum as the working electrode to eliminate any potential interference from carbon and the CMC binder.
Author Response
Replies to Reviewer 2
In this manuscript, the authors explore the application of novel ionic liquids for high-temperature lithium batteries. The study thoroughly analyzes key properties, including melting point, ionic conductivity, and both thermal and electrochemical stability. Preliminary battery performance results indicate compatibility with commercial cathode materials. This work offers valuable insights into the potential use of ionic liquids as electrolytes for next-generation lithium batteries. I recommend the manuscript for publication after addressing the following concerns. Detailed comments are provided below.
Authors’ reply
The authors are grateful to the reviewer for the positive comments devoted to improving the manuscript quality. Changes in the text are highlighted in yellow.
In Figure 2, the exothermic notation on the y-axis is misleading, as both endothermic peaks are observed for both IL and IL-LiTFSI.
Authors’ reply
The authors agree with the reviewer. The Y-axis notation of Figure 2 was modified.
Although the ionic liquid systems show high ionic conductivity under high temperatures, one potential concern is the low Li transference number, which limits the Li+ transfer in batteries. It is recommended to provide a brief comment on the ion selectivity in IL systems. Additionally, is LiFSI salt a better choice considering the weaker binding of the FSI-?
Authors’ reply
As the reviewer correctly stated, lithium batteries are affected by low Li+ transference number, generally around 0.20. This is, however, out of the purpose of the present work and it will be object of near-future work. Nevertheless, preliminary tests in lithium cells revealed that above 94 % of the theoretical capacity is still delivered at medium rates (0.5C), i.e., indicating the Li+ cation can sufficiently move through the electrolyte medium.
LiFSI does not represent a better choice because of the relatively low thermal stability of the FSI anion due to weakness of the S-F chemical bound (i.e., with respect to the C-F one of the TFSI anion).
The EMITFSI system shows the lowest electrochemical stability, which appears inconsistent with previous studies. I suggest the authors use aluminium as the working electrode to eliminate any potential interference from carbon and the CMC binder.
Authors’ reply
Inert working electrodes (such as aluminium, platinum, nickel), even if reported in literature, represent an ideal situation and do not reproduce the behaviour in a real device, i.e., they can lead to overestimation of the electrochemical stability. As the reviewer correctly stated, the battery electrodes contain carbon and binder, which affect the electrochemical behaviour of the electrolyte and, consequently, the cell performance. Therefore, aiming of obtaining more reliable results, the electrochemical stability must be investigated in carbon- and binder-containing working electrodes, which allow to much better simulate the behaviour in a practical device. The present comment was added in the manuscript.
Round 2
Reviewer 1 Report
Comments and Suggestions for Authors
Three out of the nine comments have not been addressed.
Comments on the Quality of English LanguagePlease check for typos.
Author Response
Replies to Reviewer 1 (2nd round)
Dear Reviewer
We have tried to reply to your comments. The manuscript was modified accordingly to your suggestions. Changes in the text, made with respect to the 1st revised draft of the manuscript, are highlighted in green.
Comment 1
Figure 2 represents the heat flow as a function of temperature. Temperature is a critical factor in Li-ion batteries. However, the figure only shows temperature variations. Why are the temperature values not indicated? Add the temperature values.
Authors reply
The temperature values were added in Figure 2.
Comment 2
Why is the experimental method in Section 3? Organizing the paper in a logical sequence will help readers understand it better. Adjust the order of the experimental methods and results accordingly.
Authors reply
The authors have followed the template provided by the Journal.
Comment 3
The conclusion should consist of both results and discussion. However, this conclusion merely lists the results, making it inadequate. Revise the conclusion based on references from previous literature.
Authors’ reply
The Conclusions were revised accordingly to Reviewer’s suggestions.
